# Elucidating the Prognostic and Therapeutic Implications of Insulin Resistance Genes in Breast Cancer: A Machine Learning-Powered Analysis

**DOI:** 10.3390/biology14050539

**Published:** 2025-05-13

**Authors:** Lengyun Wei, Dashuai Li, Hongjin Chen, Yajing Pu, Qun Wang, Jintao Li, Meng Zhou, Chenfeng Liu, Pengpeng Long

**Affiliations:** School of Life Science, Anhui Medical University, Hefei 230032, China; weilengyun@ahmu.edu.cn (L.W.);

**Keywords:** breast cancer, insulin resistance, risk score, overall survival, prognosis

## Abstract

The study investigated the association between insulin resistance and breast cancer through the application of various machine learning techniques. By analyzing data from a large patient cohort, seven key genes (LIFR, EZR, TBC1D4, NSF, RPL5, SAA1, and PGK1) were identified. Based on these genes, a prognostic model was constructed, which demonstrated a strong correlation with overall survival, clinical outcomes, and treatment responses. These findings suggest that the proposed model may serve as a valuable tool for clinicians in predicting treatment responses, thereby facilitating more personalized therapeutic strategies.

## 1. Introduction

Breast cancer (BC) is the most frequently diagnosed cancer and the second leading cause of cancer-related death among women worldwide [1,2]. Although adjuvant chem-otherapy and hormonal therapies have significantly reduced BC mortality, accurately pre-dicting individual patient prognosis, identifying those likely to benefit from specific treat-ments, and recognizing individuals at risk of adverse reactions remains a challenge [3]. Therefore, the discovery of novel and reliable prognostic biomarkers, as well as tools for monitoring treatment response, is of critical importance in improving outcomes for BC patients [4]. 

Insulin resistance (IR), defined as a systemic condition in which cells and tissues exhibit reduced responsiveness to the hormone insulin, affects multiple organs and regulatory pathways involved in metabolic homeostasis [5]. Numerous studies have documented a strong link between IR, visceral adiposity dysfunction, and systemic inflammation, all of which contribute to a carcinogenic environment [6]. Moreover, therapeutic strategies targeting key genes involved in IR have shown promising therapeutic effects in both preclinical models and clinical settings [7]. Recent evidence increasingly indicates that IR plays a significant role in the development and progression of breast cancer (BC) [8,9].

In this study, we developed a robust prognostic signature based on insulin resistance-related genes (IRGs) within a training cohort of BC patients. The prognostic value of this IR-based signature was subsequently validated across multiple independent cohorts. We further examined the correlation between the derived risk scores and various clinical parameters, including treatment outcomes, BC molecular subtypes, and immune checkpoint expressions. Furthermore, the predictive capacity of the IR signature was assessed in BC patients receiving chemotherapy, providing insight into its potential application in personalized treatment planning.

## 2. Materials and Methods

### 2.1. Data Collection and Processing

Transcriptome expression data and corresponding clinicopathological data for BC samples were obtained from the Cancer Genome Atlas (TCGA) and the Molecular Taxonomy of BC International Consortium (METABRIC). The distribution of major clinical features of TCGA-BRCA and METABRIC were visualized (Appendix A). Additionally, eight BC cohorts were collected from the Gene Expression Omnibus (GEO) and the Single Cell Portal. After excluding patients lacking overall survival (OS) data, a comprehensive dataset comprising 6907 BC samples across 10 cohorts was compiled (Table 1). For the RNA sequencing data, fragments per kilobase of transcript per million mapped reads (FPKM) values were transformed into transcripts per million (TPM) values. Both RNA sequencing and microarray datasets underwent log_2_ transformation and Z-score normalization to normalize the data distribution.

The training cohort for the prognostic model included 1095 BC patients from TCGA-BRCA. The validate cohort consisted of 5500 BC patients from four cohorts (METABRIC, GSE96058, GSE20685, and GSE7390). To evaluate the prognostic model’s predictive accuracy regarding therapeutic response, four additional BC cohorts (GSE191127, GSE20181, GSE18728, and GSE225078) consisting of 307 patients treated with neoadjuvant therapy were analyzed.

Insulin resistance-related genes (IRGs) were retrieved from the GeneCards database using the keyword “insulin resistance” and default search parameters. Only protein-coding genes with a relevance score greater than 3.0 were included in further analysis (Appendix A).

### 2.2. Construction and Validation of an IRG Prognostic Signature for Breast Cancer Patients

To identify significant differentially expressed genes (DEGs) between breast cancer (BC) samples and normal controls, the TCGA-BRCA cohort were analyzed using the R package limma (version 4.1.0). DEGs were identified by a threshold of absolute log2 fold change (FC) > 0.5 and an adjusted *p* value < 0.01. Among these DEGs, insulin resistance-related genes (IRGs) were subjected to univariate Cox regression analysis in both the TCGA-BRCA and METABRIC cohorts (*p* value < 0.01 in each). The overlapping significant IRGs from both cohorts were selected to develop a prognostic model. Feature selection was conducted using least absolute shrinkage and selection operator (LASSO) Cox regression analysis, a widely applied method for penalized regression and variable selection. This process identified seven hub genes with non-zero coefficients for inclusion in the prognostic model. The model calculates an insulin resistance risk score (IRRS) for each BC patient using the formula: $IRRS=\sum E_{i}*\beta_{i}$, where $E_{i}$ represents the expression level of the *i*th hub gene, and $\beta_{i}$ is its corresponding LASSO regression coefficient. The expression levels of 7 hub genes in TCGA-BRCA were visualized using a heatmap generated by the R package ggplot2 (version 4.1.3) [20].

The BC patients were stratified into two groups based on the median IRRS value. To assess survival outcomes across these groups, Kaplan–Meier (K-M) survival analyses were conducted to compare overall survival (OS) between the two groups in both the training and validation cohorts. In the training cohort, additional K-M analyses were performed to evaluate differences in disease-free survival (DFS), disease-specific survival (DSS), progression-free interval (PFI), and progression-free survival (PFS) between the risk groups.

### 2.3. Development and Evaluation of an Insulin Resistance-Related Clinicopathological Nomogram

A clinicopathological nomogram was developed by integrating IRRS with other key clinical variables, including T stage, M stage, overall stage, and PAM50 molecular subtype. Univariate and multivariate Cox regression analyses were conducted for variable selection, resulting in the retention of two significant predictors—IRRS and PAM50 (*p* < 0.05)—for inclusion in the final model. The regression coefficients from the final model were converted into a point-based scoring system using the R package rms. A graphical nomogram was constructed using the R package regplot (version 1.1), with scales for each predictor and a total points scale to estimate 1-year, 3-year, 5-year, and 10-year survival probabilities. Calibration curve analysis and decision curve analysis (DCA) were performed to confirm the satisfactory predictive discrimination of the nomogram.

### 2.4. Tumor Microenvironment Characterization Analysis

To evaluate the tumor microenvironment (TME), including its stromal and immune components, the R package ESTIMATE (version 1.0.13) [21] was applied to the TCGA-BRCA dataset. This package calculated tumor purity, immune scores, and stromal scores, providing insight into the extent of immune and stromal cell infiltration in each breast cancer (BC) sample. Additionally, CIBERSORT was used to deconvolute the gene expression profiles and estimate the relative abundance of 22 immune cell types under default parameters [22]. The distribution of these immune cell populations was compared between high- and low-IRRS groups was investigated. Furthermore, four major immune cell categories—total lymphocytes, dendritic cells, macrophages, and mast cells—were analyzed to access their relative abundance across IRRS groups.

### 2.5. Differential Gene Expression of Low- and High-IRRS Groups and Functional Enrichment Analysis

Differential gene expression analysis between the high- and low-IRRS patient groups within the TCGA-BRCA cohort was conducted using the R package limma (version 4.1.0). Gene set enrichment analysis (GSEA) was performed using the R package clusterProfiler (version 4.2.2) to identify pathways significantly associated with IRRS groups. Additionally, Gene Ontology (GO) [23] and Kyoto Encyclopedia of Genes and Genomes (KEGG) term [24,25,26] enrichment analyses were carried out through the gseGO and gseKEGG functions in the R package clusterProfiler (version 4.2.2). The gseaplot2 function from the R package enrichplot (version 1.14.2) was used to visualization the GSEA results.

### 2.6. Single-Cell RNA Sequencing Data Analysis

The raw count matrix and associated metadata that for five BC samples (SCP1106 [19]) were downloaded from the Single Cell Portal (https://singlecell.broadinstitute.org/single_cell, accessed on 21 March 2024). All samples subjected to single-cell RNA sequencing originated from triple-negative breast cancer (TNBC) cases. Low quality cells were filtered out based on the following criteria: (1) fewer than 200 unique molecular identifiers (UMIs); (2) more than 6000 or fewer than 100 expressed genes; and (3) the percentage of mitochondrial UMI being less than 10%. Subsequently, the gene expression matrices were normalized using the Normalization function, and variable gene features were identified using the FindVariableFeatures function with default parameters. To integrate data across samples, the RunFastMNN function was applied following the developer’s guidelines [27]. The top 15 principal components were used for subspace alignment and dimensionality reduction via uniform manifold approximation and projection (UMAP). Cell type annotation was conducted as previously described [19], and the analysis of single-cell RNA sequencing data was carried out using the R package Seurat (version 4.1) [28].

### 2.7. Ucell Analysis

The R package Ucell (version 2.0) was utilized to assess the enrichment of the IR signature at the single-cell level. An IR Ucell signature score, representing the enrichment of the seven hub genes in each cell, was calculated using the AddModuleScore_UCell function. Subsequently, cells were stratified into high and low groups based on their UCell scores. GSEA using a hallmark gene set was conducted to identify DEGs between the high- and low-Ucell-score groups, using the R package clusterProfiler (version 4.2.2). Normalized enrichment scores were visualized by R package ggplot2 (version 4.1.3), and Ucell score distributions were illustrated via violin plots.

### 2.8. Construction and Validation of the Machine Learning Model

The BC patients from TCGA-BRCA were randomly divided into training (70%) and testing (30%) sets using the sample.split function in the R package caTools. Pivotal IRGs involved in the prognostic model were taken as the initial features, and all data were standardized using z-score normalization. To develop a robust machine learning model, two algorithms—XGBoost and SVM—were employed.

For the XGBoost model, recursive feature elimination (RFE) with a preliminary XGBoost model was employed to refine the feature set. The final model was configured with the following parameters: (1) a learning rate (eta) of 0.4, (2) a maximum depth of trees of 2, (3) 50 boosting rounds, and (4) a binary objective function. Hyperparameter optimization was performed using grid search with 5-fold cross-validation, and the optimal parameters were selected based on the highest average ROC-AUC score. The model was trained using the xgb.train function in the R package xgboost, and feature importance was identified using xgb.importance function.

In parallel, an SVM model was built using the svm function from the R package e1071. The radial basis function (RBF) kernel was used, and hyperparameters including the cost (C) and kernel coefficient (gamma) were optimized via grid search with 5-fold cross-validation, based on ROC-AUC performance. The SVM model was trained on the same set of features selected through RFE to ensure comparability with the XGBoost model.

External validation of the XGBoost and SVM model was performed using the METABRIC and GSE96058 cohorts. Expression data for the final feature set in these validation cohorts were standardized using z-score normalization and used to predict patient groupings via the trained XGBoost model. All model training and validation procedures were executed on a system equipped with an Intel i7 processor and 32 GB of RAM.

### 2.9. Patients and Specimens

Formalin-fixed, paraffin-embedded (FFPE) tissue sections from 10 patients for immunohistochemistry (IHC) staining were obtained from the First People’s Hospital of Changzhou. Clinical samples included ER-positive, HER2-positive, and triple-negative cases across different tumor stages (I-III).

### 2.10. Immunohistochemistry

Immunohistochemistry (IHC) staining was performed as described previously [29]. Primary antibodies against CD163, CD8α, and PGK1 (Zen-Bioscience Co., Ltd., Chengdu, China) were used in this study. All primary antibodies were incubated at 4 °C overnight, followed by incubation with a horseradish peroxidase-conjugated anti-rabbit secondary antibody. All sections were visualized using a DAB substrate. For imaging, slides were scanned with a Pannoramic 250 Flash III scanner (3DHISTECH Ltd, Budapest, Hungary), and images were obtained using Pannoramic Viewer 1.1 Software (3DHISTECH Ltd.).

### 2.11. Statistical Analysis

All statistical analyses were performed using R software (Version 4.1.0). Kaplan–Meier survival curves and univariate Cox proportional hazard regression analysis were conducted with the R packages survival (version 3.7.0) and survivalminer (version 3.4.0). The LASSO algorithm was executed using the R package glmnet (version 4.1.8). Multivariate binary logistic regression, nomogram construction, and calibration curve plotting were performed using the R package rms (version 7.0.0). Differences in IRG expression levels between normal and tumor tissues, as well as changes in IRRS before and after therapy, were assessed using Student’s *t*-test. Additionally, associations between IRRS and clinicopathological features—including PAM50 subtypes, AJCC staging (T and N stages), and lymph node status—were evaluated using the Wilcoxon rank-sum test.

## 3. Results

### 3.1. Construction and Validation of an Insulin Resistance-Relevant Prognostic Signature for Breast Cancer Patients

Initially, 2828 insulin resistance-related genes (IRGs) were retrieved from GeneCards after data processing. Differential expression analysis comparing 1095 BC samples with 113 normal breast samples in the TCGA-BRCA cohort, revealed 6324 significant differentially expressed genes (DEGs). Among these, 1115 genes were recognized as significant IRGs and retained for further analysis. Univariate Cox hazard regression analysis then isolated 54 and 290 significant IRGs within the TCGA-BRCA and METABRIC cohorts, respectively. A subsequent LASSO Cox regression analysis of the 24 IRGs common to both cohorts yielded a seven-gene prognostic signature comprising LIFR, EZR, TBC1D4, NSF, RPL5, SAA1, and PGK1 (Figure 1A,B). The expression levels of these genes, correlated with patient overall survival (OS), were visualized (Appendix A). Using the established prognostic model, an insulin-related risk score (IRRS) was calculated for each patient by applying the following formula: IRRS = (0.040 × expression of EZR) + (−0.046 × expression of LIFR) + (−0.138 × expression of TBC1D4) + (−0.0105 × expression of SAA1) + (0.0218 × expression of NSF) + (−0.0566 × expression of RPL5) + (0.464 expression of PGK1). Patients were subsequently classified into low- or high-IRRS groups, based on the median IRRS value. A forest plot illustrating the relationship between gene expressions and risk of BC (Figure 1C). The analysis of hazard ratios within this plot identified the EZR, NSF, and PGK1 genes as prognostic markers in BC, with elevated expression (highlighted in red) linked to poorer outcomes. Notably, the combined IRRS factor demonstrated a significantly higher hazard ratio (HR = 4.368, 95% CI: 2.810–6.792; *p* value < 0.0001) when compared to any individual gene. A heatmap analysis further demonstrated the correlation between the expression levels of each IRG and clinical parameters within the TCGA-BRCA and METABRIC cohorts (Figure 1D, Appendix A).

To evaluate the prognostic value of the IRRS model, Kaplan–Meier (K-M) survival analysis was employed to compare overall survival (OS) between the low- and high-IRRS groups across the training cohort and four independent validation cohorts (Figure 1E–I). In all datasets, patients classified in the high-IRRS group exhibited significantly reduced OS compared to those in the low-IRRS group, as evidenced by log-rank test results (TCGA: *p* < 0.0001; METABRIC: *p* < 0.048; GSE20685: *p* = 0.012; GSE96058: *p* = 1.8 × 10^−11^; GSE7390: *p* = 0.027). Further validation of the IRRS model’s robustness was conducted through its performance assessment in predicting disease-specific survival (DSS), disease-free interval (DFI), progression-free interval (PFI), and progression-free survival (PFS) within the TCGA-BRCA cohort. Consistently, high IRRS was significantly associated with poorer survival outcomes (DSS of TCGA: *p* value = 1.4 × 10^−4^; DFI of TCGA: *p* value = 5.8 × 10^−3^; PFI of TCGA: *p* value = 8.1 × 10^−5^; PFS of TCGA: *p* value < 8.1 × 10^−0.5^; log-rank test), as demonstrated in Appendix A. In addition, the association between IRRS and overall survival (OS) across different breast cancer subtypes was explored in Appendix A. Collectively, these findings underscore the IRRS model as a significant predictive marker for survival outcomes in BC patients.

### 3.2. Integrated Assessment of the Prognostic Model and Clinical Parameters in Patients with Breast Cancer

To further evaluate the clinical relevance of the IRRS model, IRRS values were computed for each patient in the TCGA-BRCA, METABRIC, and GSE96058 cohorts, and their associations with key clinical parameters were analyzed. In the training cohort, a significant association was observed between the IRRS and critical clinical outcomes, such as PAM50 subtype, survival status, tumor stage, and clinicopathological T stage. Notably, an increase in the IRRS corresponded to a worsening of these clinical indicators (Figure 2A). These findings were validated in the METABRIC cohort, where IRRS was significantly associated with PAM50 subtype, tumor stage, tumor size, and tumor grade (Figure 2B). Similar trends were observed in the GSE96058 cohort, where higher IRRS values were predictive of more aggressive PAM50 subtypes, shorter survival, larger tumor size, and greater lymph node involvement (Figure 2C). Collectively, these findings highlight the strong alignment between IRRS and established clinical parameters, indicating that higher IRRS values are associated with adverse clinical outcomes in BC patients.

### 3.3. Development and Evaluation of an Insulin Resistance-Related Clinicopathologic Nomogram

To enhance individualized prognostication, nomograms incorporating both IRRS and PAM50 subtype were developed for BC samples within the TCGA-BRCA and METABRIC cohorts [30]. These nomograms were designed to estimate 1-, 3-, 5-, and 10-year overall survival (OS) for individual patients. The score for each factor reflects its respective contribution to survival outcome prediction, with results indicating a more significant contribution from IRRS compared to PAM50 subtype (Figure 3A, Appendix A). To evaluate the predictive accuracy of the nomograms, calibration curves were generated by comparing predicted survival probabilities with actual outcomes. Except for the 10-year OS predictions, the calibration curves showed good agreement with the ideal reference line for both the TCGA-BRCA and METABRIC cohorts (Figure 3B–E, Appendix A). Furthermore, decision curve analysis for 1-, 3-, 5-, and 10-year survival demonstrated that the nomograms offer a greater net benefit over standard treat-none or treat-all approaches (Figure 3F). The predictive performance of the nomogram was further quantified using the concordance index (C-index), which was 0.64 in the TCGA-BRCA cohort and 0.59 in the METABRIC cohort. Notably, our IRRS-based gene signature outperformed several previously published breast cancer biomarkers, as evidenced by its superior C-index values (Table 2).

### 3.4. Association of TME Subcomponents with IRRS and Outcome in Patients with Breast Cancer

The tumor microenvironment (TME) plays a critical role in cancer progression and significantly impacts patient survival outcomes [36]. To explore differences in TME composition between high- and low-IRRS groups, we utilized the ESTIMATE algorithm to quantify stromal and immune components. As shown in Figure 4A, the tumor purity of the low-IRRS group was significantly lower compared to the high-IRRS group, corroborating the observation that higher IRRS is associated with poorer survival outcomes. This observation supports the association between elevated IRRS and poorer prognosis. Furthermore, the results demonstrated that the low-IRRS group had markedly higher immune and stromal scores than their high-IRRS counterparts.

To further characterize immune infiltration differences between low- and high-IRRS groups, we analyzed the distribution of immune cell types across both groups. Figure 4B presents the relative proportions of immune cell populations, while Figure 4C displays these distributions as boxplots. A comparative analysis comparing the proportions of each cell type between the groups is shown in Figure 4D. Notably, the high-IRRS group exhibited significantly higher proportions of M0 macrophages, M2 macrophages, neutrophils, resting NK cells, and activated memory CD4+ T cells compared to the low-IRRS group. Conversely, the low-IRRS group had higher proportions of memory B cells, naive B cells, resting dendritic cells, resting mast cells, monocytes, activated NK cells, resting memory CD4+ T cells, and CD8 T cells (*p* value < 0.05; Wilcoxon test).

Additionally, to provide a broader perspective, four aggregated immune cell categories were also compared between groups, including total lymphocytes, total dendritic cells (sum of activated and resting), total macrophages (sum of M0, M1 and M2), and total mast cells (sum of activated and resting) between the two groups, as illustrated in Figure 4E. The analysis revealed that the low-IRRS group had significantly higher levels of total dendritic cells, total lymphocytes, and total mast cells, while the high-IRRS group had a greater total macrophage count. Furthermore, gene expression analysis of the TCGA-BRCA dataset indicated a significant reduction in CD8a expression within the high-IRRS group (Figure 4F), suggesting a diminished cytotoxic CD8+ T cell response, which is critical for anti-tumor immunity. Conversely, CD163 expression, a marker of immunosuppressive M2 macrophages, was significantly upregulated in the high-IRRS group (Figure 4G), indicating an immune-evasive microenvironment (Figure 4G). Immunohistochemistry (IHC) staining on human BC tissues for CD8a, CD163, and PGK1 further detailed the correlation between PGK1 expression and immune cell infiltration. High PGK1-expressing tumors were infiltrated more by CD163+ macrophages and less by CD8+ T cells. Contrastingly, low PGK1 expression areas showed increased CD8+ T cell infiltration and fewer CD163+ macrophages (Figure 4H). This inverse relationship between PGK1 expression and CD8+ T/CD163+ cell infiltration suggests that PGK1 may play a role in modifying the BC immune environment, indicating a significant immunosuppressive TME in the high-IRRS group that may promote immune escape and worsen BC prognosis. These findings highlight a strong immunosuppressive tumor microenvironment (TME) in the high-IRRS group, potentially facilitating tumor cell immune escape and leading to a poorer prognosis in BC patients.

### 3.5. Bioinformatic Analysis of the Characteristics and Signaling Pathways Among Patients in Different Risk Groups

To elucidate the biological distinctions between the low- and high-IRRS groups, we conducted a comprehensive analysis of signaling pathways enriched among differentially expressed genes (DEGs). Initially, 1470 significant DEGs were pinpointed using criteria of absolute log2FC > 1 and FDR < 0.01, of which 501 were upregulated and 969 were downregulated in BC samples (Figure 5A). KEGG pathway analysis indicated that the high-IRRS group was associated with pathways including PI3k-Akt signaling, cell cycle, focal adhesion, extracellular matrix (ECM)-receptor interaction, HIF-1 signaling, PPAR signaling, and regulation of lipolysis in adipocytes (Figure 5B). GO analysis focusing on the biological process (BP) highlighted nuclear division as a primary activity in the high-IRRS group, while cellular component (CC) analysis showed enrichment in collagen-containing ECM. Analysis of molecular function (MF) revealed a significant role in signaling receptor activator activity (Figure 5C). Further, gene set enrichment analysis (GSEA) of GO, KEGG, and hallmark gene sets revealed additional pathways significantly enriched in the high-IRRS group, including chromosome segregation, DNA replication, endothelial cell proliferation, and diverse metabolic processes (Figure 5D). Specifically, GSEA of KEGG terms indicated that the high-IRRS group activated pathways like nucleotide sugars biosynthesis, cell cycle, and DNA replication (Figure 5E), along with hallmark pathways such as adipogenesis, bile acid metabolism, and glycolysis (Figure 5F). To further investigate the relationship between the high-IRRS group and insulin resistance, we performed GSEA on the insulin-related pathway. As shown in Appendix A, the pathways, including insulin signaling pathway, response to insulin, and PI3K-AKT signaling, were down-regulated in the high-IRRS group. Together, these findings highlight the distinct molecular landscapes of the low- and high-IRRS groups. The high-IRRS subtype is characterized by upregulation of cell proliferation, metabolic reprogramming, and ECM remodeling pathways, underscoring the complex and multifaceted biological processes that may drive poorer prognosis in these patients.

### 3.6. Therapeutic Benefit of the IRG Prognostic Signature

To evaluate the predictive utility of the IRG-based prognostic signature in the context of neoadjuvant chemotherapy, we analyzed pre- and post-treatment IRRS values across four breast cancer (BC) cohorts: GSE191127, GSE20181, GSE18728, and GSE225078. In the GSE191127 cohort, pretreatment IRRS was elevated compared to post-treatment IRRS (Figure 6A, *p* value = 1.8 × 10^−2^; T test), indicating a reduction in IRRS following chemotherapy. Additionally, comparisons between responders and nonresponders, as well as between recurrent and nonrecurrent cases, confirmed that lower IRRS values were associated with better treatment response and reduced recurrence risk (Figure 6B,C, nonresponse vs. response: *p* value = 3.8 × 10^−2^; nonrecurrent vs. recurrent: *p* value = 1.5 × 10^−3^; T test). In the GSE20181 cohort, significant IRRS reduction was observed when comparing prechemotherapy levels with those on Day 14 and Day 90 post-treatment (Figure 6D,E, *p* value = 4.3 × 10^−1^ for before vs. Day 14; *p* value = 3.2 × 10^−4^ for before vs. Day 90; T test), with the 14-day treatment group displaying a notably higher IRRS than the 90-day treatment group (Figure 6F, *p* value = 1.7 × 10^−5^ for Day 14 vs. Day 90; T test). In the GSE18728 cohort, a significant reduction in IRRS was also observed after one cycle of adjuvant chemotherapy (C1) and in the definitive surgical specimen (OR) among patients treated with docetaxel (T) and capecitabine (X) (Figure 6G,H, *p* value = 2.3 × 10^−2^ for C1 vs. BL; *p* value = 6.9 × 10^−3^ for OR vs. BL; T test). Similarly, in the GSE225078 cohort, patients treated with pembrolizumab plus paclitaxel exhibited a substantial decrease in IRRS compared to prechemotherapy levels (Figure 6I, *p* value = 1.5 × 10^−3^). These findings collectively demonstrate that a lower IRRS is associated with increased therapeutic efficacy, suggesting that IRRS may serve as a useful biomarker for predicting responses to neoadjuvant chemotherapy in BC patients.

### 3.7. Single-Cell RNA Sequencing Data Analysis

To further dissect the cellular landscape underlying the IRRS signature, we performed single-cell RNA sequencing (scRNA-seq) analysis on five breast cancer samples, resulting in 24,271 high-quality cells after rigorous quality control. Unsupervised clustering revealed 19 transcriptionally distinct clusters (Figure 7A). Additionally, cell type annotation identified eight major cell populations: T cells, myeloid cells, epithelial cells, plasma cells, B cells, cancer-associated fibroblasts (CAFs), endothelial cells, and perivascular-like cells (PVLs) (Figure 7B). The expression levels of the seven IRRS hub genes across each cell type are illustrated in Figure 7C,D. To quantify the activity of these genes at the single-cell level, an IR Ucell score was calculated for each cell based on seven hub genes using the R package UCell (Figure 7E). Subsequent analysis divided the cells into two categories based on the IR Ucell score: top 25% (high IR Ucell score) and bottom 25% (low IR Ucell Score), as demonstrated in Figure 7F. Interestingly, the high-IR-Ucell-score group showed a higher proportion of T cells, epithelial cells, and B cells compared to the low-score group (Figure 7G). Furthermore, GSEA based on the hallmark gene set revealed that, within epithelial clusters, high-IR-Ucell-score cells exhibited significant enrichment of cancer-associated pathways such as epithelial homeostasis and the unfolded protein response (Figure 7H). These single-cell data reinforce the relevance of the IRRS signature in defining functional heterogeneity within the tumor microenvironment and highlight its potential role in shaping immune and epithelial cell dynamics in breast cancer.

### 3.8. Prediction of the Low- and High-IRRS Subtypes by the XGBoost Algorithm

To develop a robust classifier for distinguishing between low- and high-IRRS groups in BC samples, we implemented and compared two machine learning algorithms: extreme gradient boosting (XGBoost) and a support vector machine (SVM). Both models utilized the seven hub genes from the IRRS signature as input features. For the XGBoost classifier, Shapley additive explanations (SHAP) were used to interpret the marginal contributions of each gene during a 10-fold cross-validation process (Figure 8A). The TCGA-BRCA cohort was randomly split into training (80%) and validation (20%) sets. SHAP analysis identified PGK1 and TBC1D4 as the most influential genes in the mode (Figure 8B). Performance evaluation of the validation set showed an overall accuracy of 0.86 and an area under the receiver operating characteristic curve (AUC) of 0.94 (Figure 8C). To assess the model’s capacity to classify new data accurately, we also analyzed the GSE96058 and METABRIC cohorts. Figure 8D presents the overall accuracy and AUC of GSE96058, which were determined to be 0.88 and 0.83, respectively. However, the overall accuracy and AUC of METABRIC, as shown in Figure 8E, were 0.74 and 0.70, respectively, and were comparatively lower than those of GSE96058.

In parallel, we constructed an SVM classifier using the R package e1071 with a radial basis function (RBF) kernel. In the TCGA-BRCA dataset, the SVM model achieved an accuracy of 0.80 and an AUC of 0.91—slightly lower than XGBoost (Appendix A). SVM also showed worse performance than XGBoost in the validation cohorts GSE96058 (0.80 for accuracy and 0.77 for AUC) and METABRIC (0.69 for accuracy and 0.74 for AUC) (Appendix A). Compared to SVM, XGBoost consistently demonstrated higher predictive accuracy across the datasets.

## 4. Discussion

Mounting evidence indicates that insulin resistance (IR) within both cancer cells and the tumor microenvironment (TME) plays a pivotal role in tumor progression, recurrence, metastasis, and resistance to various therapies, including chemotherapy, endocrine therapy, and radiotherapy [37,38,39,40,41]. However, the critical molecular markers linking IR to breast cancer (BC) outcomes remain insufficiently characterized. In this study, we developed and validated a prognostic signature based on seven insulin resistance-related genes (IRGs) that is predictive of overall survival (OS) and therapeutic response in BC patients.

Among the seven IRGs, several have been reported in previous studies [42,43,44]. These IRGs not only play a role in regulating insulin levels but also partake in the intricate regulation of BC progression, particularly influencing cell proliferation and metastasis [45,46,47]. For instance, studies have shown that inhibition of PGK1 (phosphoglycerate kinase 1) phosphorylation suppresses glycolysis, tumorigenesis, and cell proliferation in various cancers [48,49]. Moreover, PGK1 silencing has been demonstrated to increase paclitaxel sensitivity in triple-negative BC through XAF1-mediated apoptosis [50]. Feedback activation of the leukemia inhibitory factor receptor (LIFR) signaling pathway in BC can limit the inhibitory response to HDAC inhibitors, marking it as a potential therapeutic target [51]. Furthermore, in vitro overexpression of ribosomal protein L5 (RPL5) has been shown to significantly inhibit BC cell proliferation, the G1-S cell cycle transition, and induce apoptosis [52]. TBC1D4 (AS160), a key regulator of GLUT4 translocation, remains underexplored in BC but may contribute to insulin-mediated metabolic alterations in tumors. Despite their individual contributions, prognostic prediction based solely on single IRG expression remains limited due to the complexity of tumor biology. Our findings emphasize that cancer progression is likely influenced by a coordinated network of genes, reinforcing the need for integrated models such as the IRRS signature.

The IRRS model effectively stratified patients by survival risk and showed strong predictive performance across multiple cohorts. Nomogram analysis further validated that the IRG signature possessed high precision in forecasting 1-year, 3-year, 5-year, and 10-year OS rates for BC patients, demonstrating its robust potential for clinical application. A primary challenge in current BC treatment is overcoming therapeutic resistance and metastasis, with many patients experiencing disease progression after initial treatment success in clinical settings [53,54]. Consequently, this study assessed the prognostic value of the IRG signature in predicting patient response to neoadjuvant chemotherapy. Findings indicated that the IRG signature was an effective biomarker for assessing treatment efficacy in various patient groups, including those pre- and post-therapy (GSE191127 and GSE225078), with differing survival outcomes (GSE20181), and those categorized by treatment response or nonresponse, and recurrence or nonrecurrence (GSE191127). Consistently, patients exhibiting a low IRRS posttreatment, those with long-term survival, and those who responded to treatment or did not experience recurrence were more likely to benefit significantly from therapeutic interventions.

Nonetheless, this study has several limitations. First, while data normalization (z-score normalization and log2 transformation) was applied to account for platform-specific differences (RNA-seq vs. microarray), the use of multiple public datasets introduces potential batch effects and clinical heterogeneity. Additionally, intratumoral and interpatient heterogeneity, known to affect treatment responses, were not fully accounted for in our analyses. Third, the single-cell RNA sequencing analysis was based on a relatively small sample size with limited genetic diversity, which may constrain the generalizability of our immune landscape findings. Lastly, the biological functions and mechanistic roles of the seven IRGs were inferred through bioinformatic analyses rather than experimentally validated. Future studies should focus on functional validation of these genes through in vitro and in vivo experiments. Investigate the potential of small molecules or biologics that specifically target the identified IRGs, screening for modulators of their expression or activity, and evaluating their therapeutic potential in preclinical models are important next steps. Additionally, assessing the synergistic effects of combining IRG-targeted therapies with existing treatment modalities, such as chemotherapy, endocrine therapy, and immunotherapy, will help determine if these combinations can enhance therapeutic outcomes.

## 5. Conclusions

In conclusion, this study identified and validated a robust prognostic signature based on insulin resistance-related genes, capable of predicting survival outcomes and therapeutic responses in breast cancer patients. This model holds promise for clinical application, potentially enhancing OS rates and facilitating the development of personalized treatment strategies incorporating targeted therapeutics.

## Figures and Tables

**Figure 1 biology-14-00539-f001:**
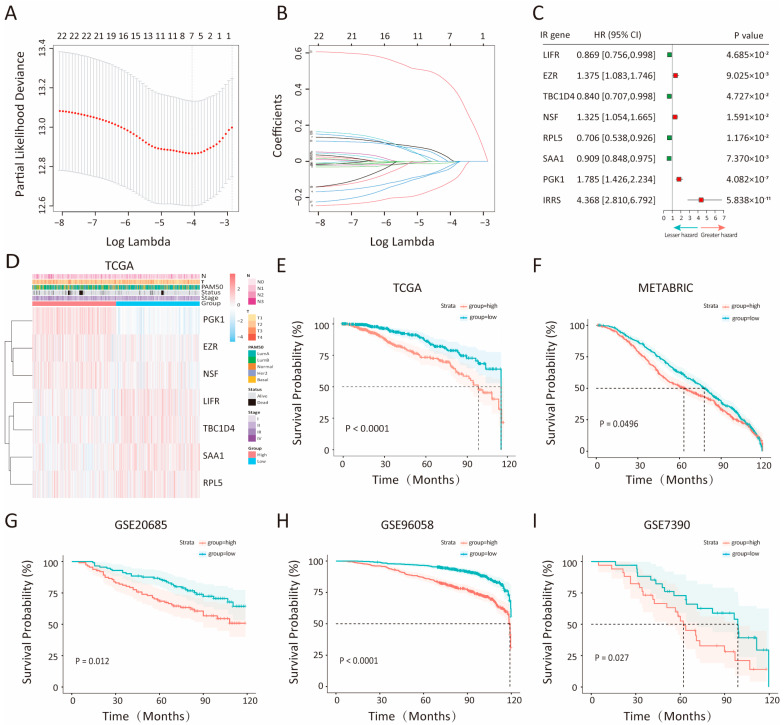
Construction and validation of the insulin resistance risk score (IRRS) model. (**A**) Partial likelihood deviance displayed by the LASSO regression model during 10-fold cross-validation. Vertical dotted lines represent optimal values determined using minimum and 1-SE criteria. (**B**) LASSO coefficient profiles of 24 insulin resistance genes selected during 10-fold cross-validation. (**C**) Multivariate Cox analysis assessing the independent predictive capability of 7 hub IRGs and IRRS for overall survival (OS). Unadjusted hazard ratios (HRs) are provided with 95% confidence intervals. A lesser hazard direction signifies better survival, while a greater hazard direction indicates lower survival. (**D**) Heatmap illustrating the expression levels of 7 hub genes in the training cohort (TCGA-BRCA). The color scale ranges from red (indicating high-IRRS group) to blue (indicating low-IRRS group) to denote magnitude. The heatmap was generated by the R package ggplot2 (version 4.1.3). (**E**–**I**) Kaplan–Meier curves for high- and low-IRRS groups in the TCGA-BRCA (**E**), METRBRIC (**F**), and cohorts GSE20658 (**G**), GSE96058 (**H**), and GSE7390 (**I**).

**Figure 2 biology-14-00539-f002:**
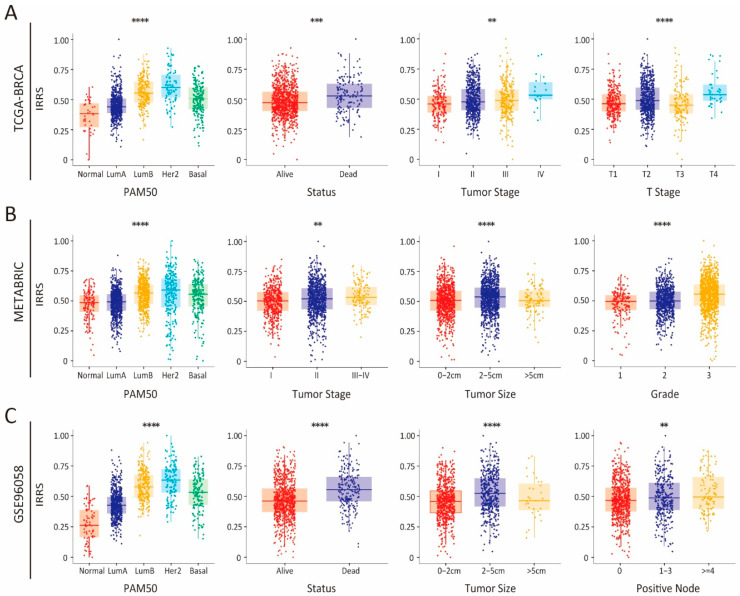
Comprehensive evaluation of the IRRS and clinical parameters in breast cancer (BC) patients. Correlation between IRRS levels and diverse clinical parameters in the TCGA-BRCA (**A**), METABRIC (**B**), and GSE96058 (**C**) cohorts. ** *p* < 0.01, *** *p* < 0.001, **** *p* < 0.0001.

**Figure 3 biology-14-00539-f003:**
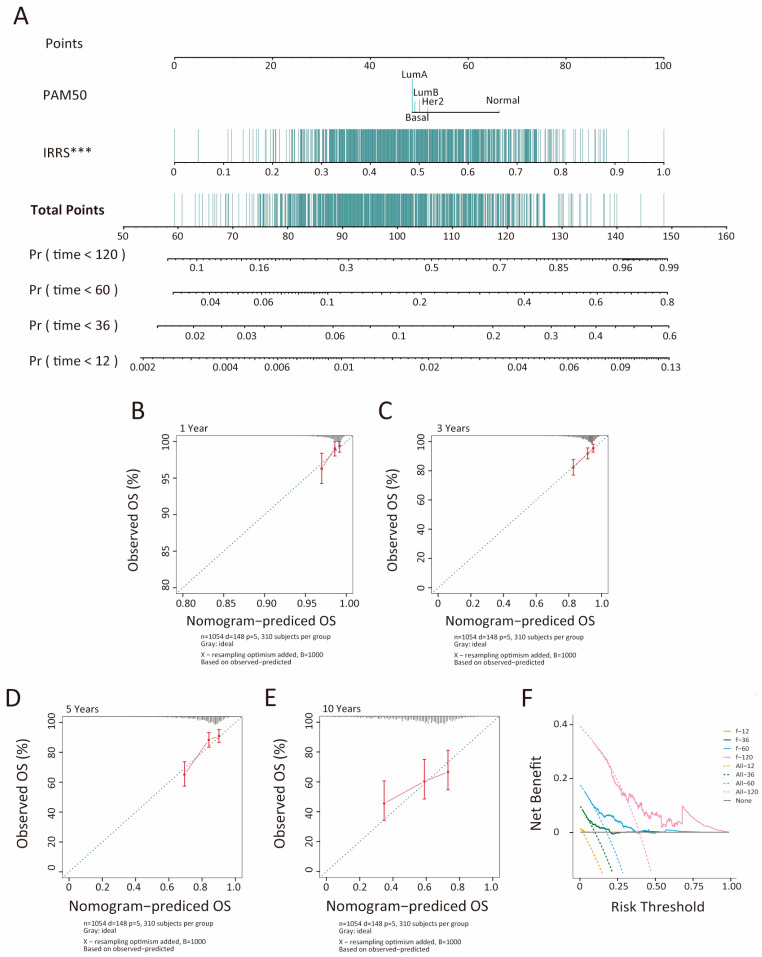
Nomogram developed to predict the probability of 1-, 3-, 5-, and 10-year overall survival in the TCGA-BRCA cohort. (**A**) Nomogram constructed for overall survival prediction in the TCGA-BRCA cohort. Calibration curves depict agreement between predicted and observed 1- (**B**), 3- (**C**), 5- (**D**), and 10-year (**E**) survival outcomes. Decision curve analysis (DCA) assesses 1-, 3-, 5-, and 10-year risks (**F**). *** *p* < 0.001.

**Figure 4 biology-14-00539-f004:**
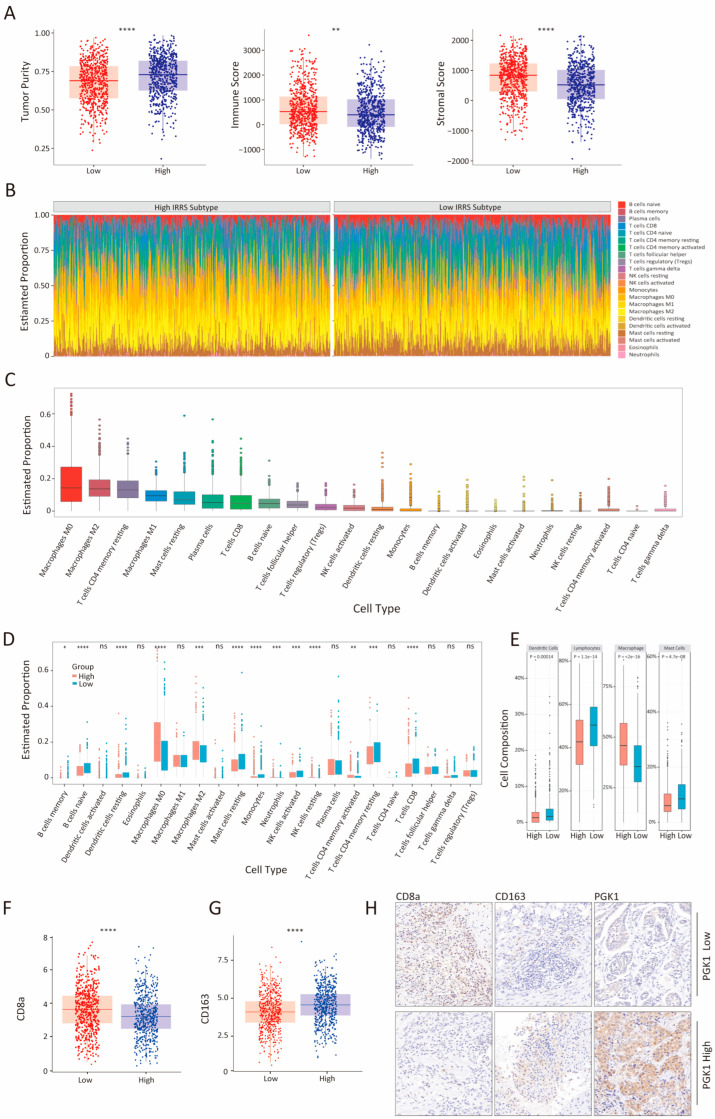
Analysis of immune cell proportions in high- and low-IRRS patients within the TCGA-BRCA cohort. (**A**) Boxplots illustrating tumor purity, immune score, and stromal score differences between low- and high-IRRS groups. (**B**) Proportions of 22 immune cell types infiltrating low- and high-IRRS groups. (**C**) Boxplot illustrating proportions of 22 immune cell types within the TCGA-BRCA cohort. (**D**) Distribution of proportions of 22 immune cell types between low- and high-IRRS groups. (**E**) Boxplots representing cell composition of total dendritic cells, total lymphocytes, total macrophages, and mast cells within the TCGA-BRCA cohort. Statistical significance: * *p* < 0.05; ** *p* < 0.01; *** *p* < 0.001; **** *p* < 0.0001; ns (not significant), *p* > 0.05. (**F**,**G**) mRNA expression level of CD8a and CD163 in low- and high-IRRS groups in TCGA. (**H**) CD8a, CD163, and PGK1 protein levels in BC tissues and adjacent normal tissues were examined by IHC. Representative images were shown (magnification at ×40). Scale bar: 20 μm.

**Figure 5 biology-14-00539-f005:**
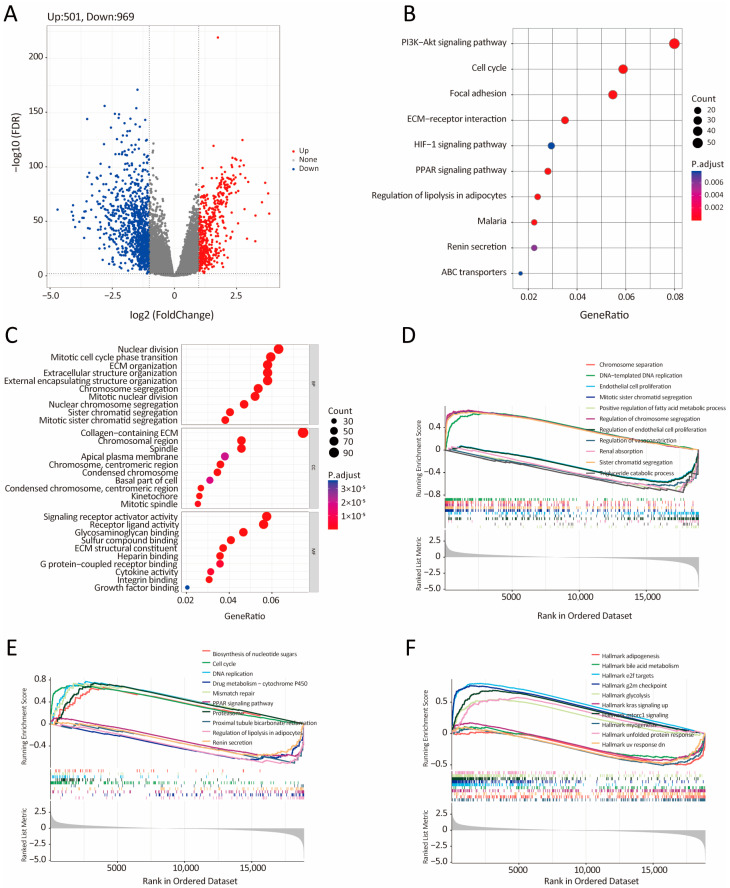
Bioinformatics analysis of characteristics and signaling pathways among patients in distinct risk groups. (**A**) Volcano plot illustrating differentially expressed genes (DEGs) in the high-IRRS group compared to the low-IRRS group within the TCGA-BRCA cohort. Genes labeled in red or green indicate significant up- or downregulation, respectively. FC: fold change; FDR: false discovery rate. (**B**) Kyoto Encyclopedia of Genes and Genomes (KEGG) [24,25,26] analysis and (**C**) Gene Ontology (GO) analysis of significant DEGs. Gene set enrichment analysis (GSEA) enriched in GO (**D**), KEGG (**E**), and hallmark (**F**) gene sets.

**Figure 6 biology-14-00539-f006:**
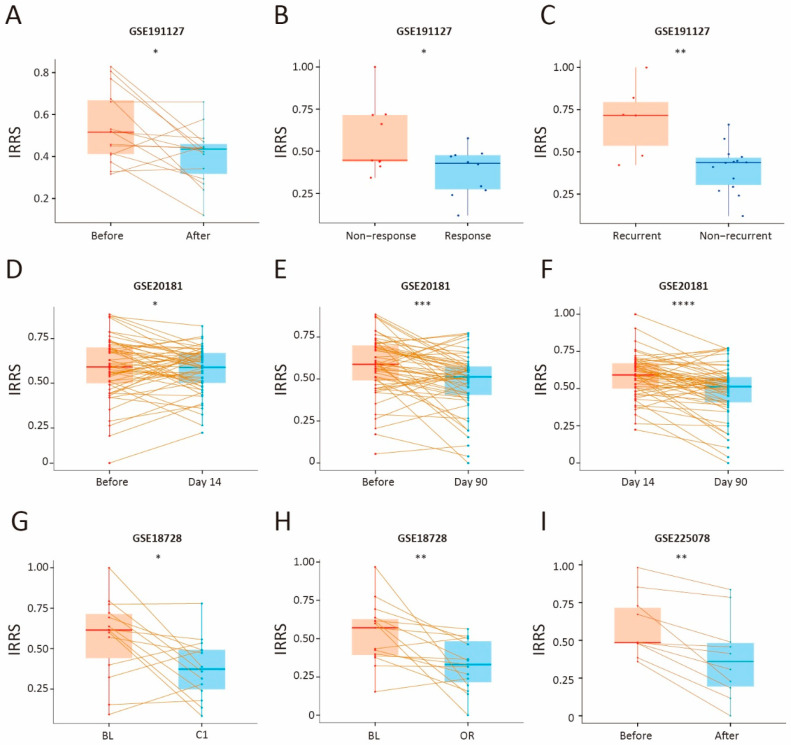
Therapeutic benefit assessment of the IRRS value. Pairwise comparison of IRRS between pre- and post-chemotherapy patients in GSE191127 (**A**–**C**), GSE20181 (**D**–**F**), GSE18728 (**G**,**H**), and GSE225078 (**I**). Significance assessed using pairwise t-test. Statistical significance denoted as * *p* < 0.05; ** *p* < 0.01; *** *p* < 0.001; **** *p* < 0.0001.

**Figure 7 biology-14-00539-f007:**
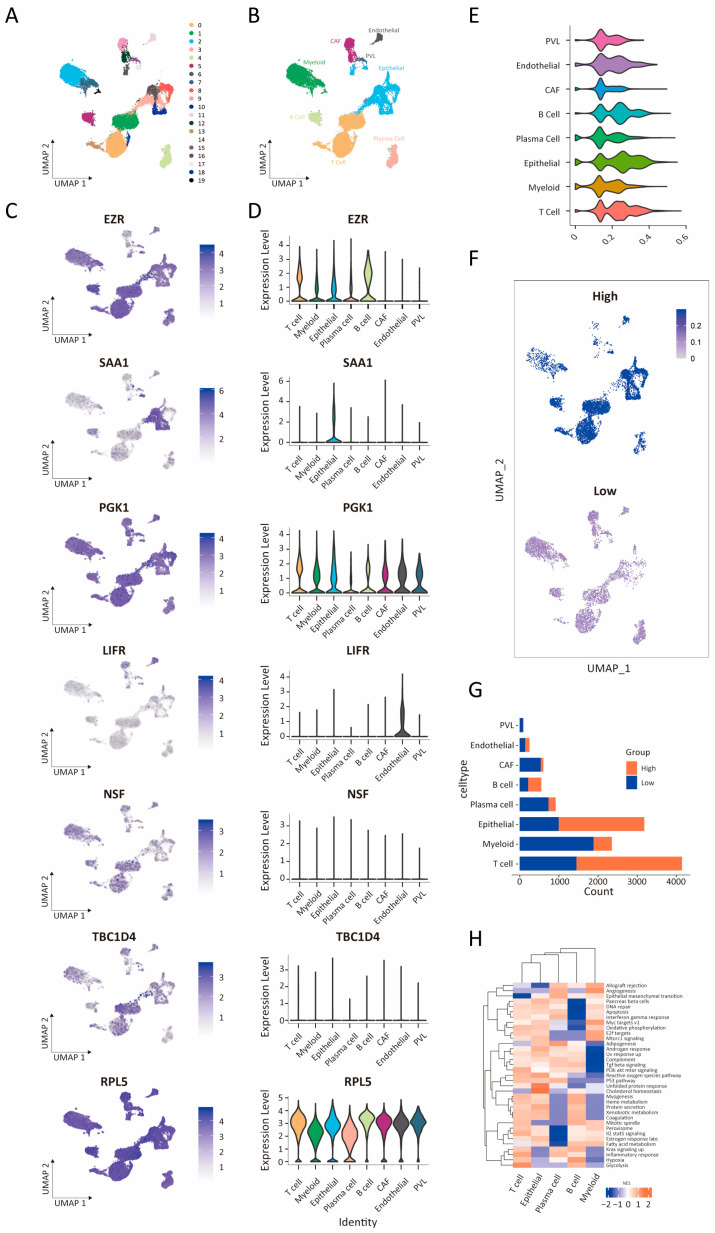
Analysis of 7 IRGs using single-cell RNA sequencing (scRNA-seq). (**A**) Clustering of cells from 5 breast cancer samples into 19 clusters. (**B**) Subclustering of aggregated scRNA-seq data from 5 breast cancer samples into T cell, myeloid cell, epithelial cell, plasma cell, B cell, cancer-associated fibroblast (CAF), endothelial cell, and perivascular-like cell (PVL). (**C**,**D**) Feature plots and violin plots displaying expression level distributions of 7 IRGs across different cell types. (**E**) Violin plot showing IR Ucell scores across various cell types. (**F**) Distribution of IR Ucell scores among cells, divided into a high-IR-Ucell-score group (top 25%) and a low-IR-Ucell-score group (bottom 25%). (**G**) Distribution of cells in the high-IR-Ucell-score or low-IR-Ucell-score groups. (**H**) Heatmap displaying pathways enriched by GSEA analysis in the hallmark gene set. The heatmap was generated by the R package ggplot2 (version 4.1.3, https://github.com/tidyverse/ggplot2, accessed on 15 May 2024).

**Figure 8 biology-14-00539-f008:**
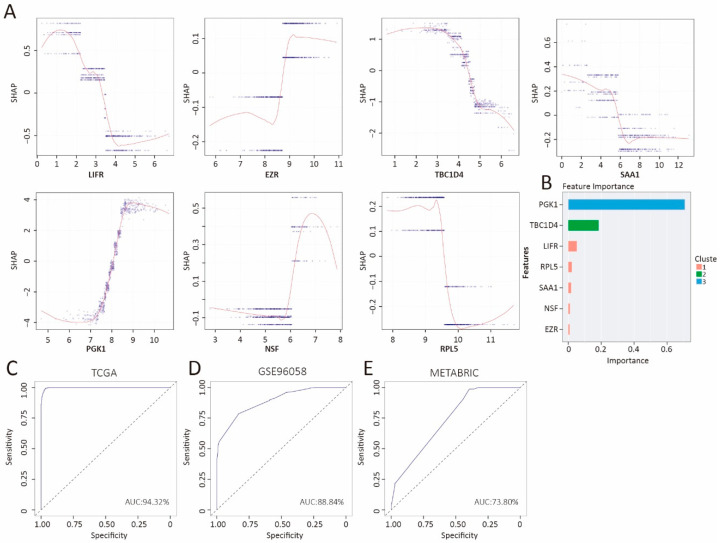
Prediction outcomes of the XGBoost algorithm. (**A**) SHAP contribution dependency plots depicting the seven crucial features. The x-axis represents the log2(TPM) of each feature. (**B**) Contributions of the seven crucial features for the TCGA-BRCA cohort. ROC curves generated by the XGBoost algorithm for predicting high- and low-IRRS subtypes in (**C**) the TCGA-BRCA, (**D**) GSE96058, and (**E**) METABRIC cohorts.

**Table 1 biology-14-00539-t001:** Cohorts involved in this study.

Cohort	Data Type	Source	Sample Size	Reference
TCGA-BRCA	RNASeq	TCGA	1095	[10]
METABRIC	microarray	cBioPortal	1906	[11]
GSE20685	microarray	GEO	327	[12]
GSE96058	RNASeq	GEO	3069	[13]
GSE7390	microarray	GEO	198	[14]
GSE191127	RNAseq	GEO	40	[15]
GSE20181	microarray	GEO	176	[16]
GSE18728	microarray	GEO	61	[17]
GSE225078	RNAseq	GEO	30	[18]
SCP1106	scRNA-seq	Single-cell portal	5	[19]

**Table 2 biology-14-00539-t002:** C-index of nomogram constructed by IRRS and published signatures.

Signature	C-Index	Reference
Yang.FG	0.56	[31]
Zhang.FG	0.53	[32]
Wang.FG	0.58	[33]
Wang.BB	0.63	[34]
IRRS	0.64	[35]

## Data Availability

All the RNA sequencing data of breast cancer were acquired from the Gene Expression Omnibus database (GEO, https://www.ncbi.nlm.nih.gov/geo/query/acc.cgi, accessed on 24 January 2024).

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
