# Peer review of "Elucidating the Prognostic and Therapeutic Implications of Insulin Resistance Genes in Breast Cancer: A Machine Learning-Powered Analysis"

_biology, 2025, doi:10.3390/biology14050539_

Round 1
Reviewer 1 Report
Comments and Suggestions for Authors
Authors presents a machine learning analysis investigating the role of insulin resistance-related genes in breast cancer. The study identified seven key genes and developed a prognostic model that predicts patient survival, clinical outcomes, and immune response. The research explores the tumor microenvironment, identifying immune cell differences between high- and low-risk groups, and assesses the model's ability to predict response to chemotherapy.
- The authors need to explore the training datasets further to characterize what is the model really learning, i.e., explore the limitations of the TCGA-BRCA dataset, and characterize the kind of samples, age, stage of cancer, etc. This will explain what is the model trained on, and is the model being tested on truly diverse data, which is of a different patient types than the one it was trained on.
- The authors do a great job in validating the signature through data, but they need to establish a mechanistic link (through literature survey) as to how this signature makes sense from biology point of view
- The authors validate using IHC of breast cancer samples. They need to talk about the diversity of samples used, and how these were enough IHC stainings to show proof of concept.
Reviewer 2 Report
Comments and Suggestions for Authors
This study developed a machine learning-based prognostic model for breast cancer using insulin resistance-related genes (IRGs). The model identified seven key genes (LIFR, EZR, TBC1D4, NSF, RPL5, SAA1, PGK1) and showed that it can predict patient survival and treatment response. I think this paper will be interesting for readers of the "Biology" journal after addressing the following comments:
- You define insulin resistance-related genes (IRGs) via GeneCards based on a relevance score > 3. How do you justify using this as a biological ground truth for “insulin resistance” in cancer, rather than a broader metabolic stress signature?
- PGK1, a glycolytic enzyme, dominates your model, but it's also a marker of general tumor proliferation. What evidence supports its specificity to insulin resistance rather than just being an approximation for aggressive tumor metabolism?
- The LASSO model has a regularization parameter as lambda. How was the optimal lambda selected, and was the sensitivity of the model to different lambda values assessed?
- Your XGBoost classifier used very shallow trees (max depth = 2) with a high learning rate (0.4). Did you benchmark this against other algorithms (such as random forest, SVM)? If not, how can you be sure your model was tuned and not limited by insufficient optimization?
- METABRIC data was collected using microarray, while TCGA and GSE96058 are RNA-seq but your model was trained and applied across both types without correction for platform bias. How did you normalize across platforms to make sure fair validation?
- How does your model perform in different molecular subtypes of breast cancer (such as triple-negative and luminal A)? Could the signature be subtype-driven rather than IR-specific?
- Only 5 single-cell samples were analyzed, but you draw conclusions about cell-type-specific enrichment of IR signature genes. Were these 5 tumors representative in subtype and stage? Could findings be driven by sample bias?
- You performed log2 transformation on gene expression data. Were other normalization methods considered, and why was log2 chosen?
- The UCell scoring approach divides cells into high/low IR gene expression. However, this is an unsupervised threshold, did you test how sensitive your results were to the percentile cutoff ?
- Have you tested whether high-IRRS tumors show evidence of insulin signaling alterations (such as AKT/mTOR activation, GLUT4 translocation)? Without this, how can you claim mechanistic ties to insulin resistance?
- You conclude that T cells and epithelial cells highly express IR genes, but isn't this expected, given that IR genes are metabolic and housekeeping in nature? What makes this enrichment meaningful in the cancer context?
- You classified patients into high vs. low IRRS post-treatment and show associations with response. But isn't this circular, since IRRS may drop simply due to reduced tumor burden, not treatment sensitivity?
Round 2
Reviewer 1 Report
Comments and Suggestions for Authors
The authors have sufficiently addressed concerns.
Reviewer 2 Report
Comments and Suggestions for Authors
The authors addressed most of my concerns and comments.